# Isolation, Identification and Genetic Characterization of Antibiotic Resistant *Escherichia coli* from Frozen Chicken Meat Obtained from Supermarkets at Dhaka City in Bangladesh

**DOI:** 10.3390/antibiotics12010041

**Published:** 2022-12-27

**Authors:** Mridha. Md. Kamal Hossain, Md. Sharifull Islam, Md. Salah Uddin, A. T. M. Mijanur Rahman, Asad Ud-Daula, Md. Ariful Islam, Rubaya Rubaya, Anjuman Ara Bhuiya, Md. Abdul Alim, Nusrat Jahan, Jinquan Li, Jahangir Alam

**Affiliations:** 1Animal Biotechnology Division, National Institute of Biotechnology, Savar 1349, Bangladesh; 2Center for Cancer Immunology, Institute of Biomedicine and Biotechnology, Shenzhen Institutes of Advanced Technology, Chinese Academy of Sciences, Shenzhen 518055, China; 3Department of Applied Nutrition and Food Technology, Islamic University, Kushtia 7003, Bangladesh; 4State Key Laboratory of Agricultural Microbiology, Huazhong Agricultural University, Wuhan 430070, China

**Keywords:** frozen chicken meat, *E. coli*, Antimicrobials, Antimicrobial resistance genes, Dhaka city

## Abstract

Antimicrobials have been used to improve animal welfare, food security, and food safety that promote the emergence, selection, and dissemination of antimicrobial-resistant (AMR) bacteria. In this study, 50 *E. coli* were isolated from frozen chicken meat samples in Dhaka city. Antibiotic sensitivity patterns were assessed through the disk diffusion method and finally screened for the presence of antimicrobial resistance genes (ARG) using the polymerase chain reaction (PCR). Among the 160 samples, the prevalence of *E. coli* was observed in fifty samples (31.25%). All of these isolates were found resistant to at least one antimicrobial agent, and 52.0% of the isolates were resistant against 4–7 different antimicrobials. High resistance was shown to tetracycline (66.0%), followed by resistance to erythromycin (42.0%), ampicillin and streptomycin (38.0%), and sulfonamide (28.0%). In addition, the most prevalent ARGs were *tet(A)* (66.0%), *ereA* (64.0%), *tet(B)* (60.0%), *aadA1* and *sulI* (56.0%), *blaCITM* (48.0%) and *blaSHV* (40.0%). About 90.0% of isolates were multidrug resistant. This study reveals for the first time the current situation of *E. coli* AMR in broilers, which is helpful for the clinical control of disease as well as for the development of policies and guidelines to reduce AMR in broilers production in Bangladesh.

## 1. Introduction

Antimicrobial agents have been used in humans, veterinary medicine, food security, and food safety since their discovery in the 1920s. However, due to inadequate selection, overuse, and misuse of antimicrobials have been responsible for the selection of resistant isolates, known as antimicrobial resistance (AMR) [1]. Over the past decade, AMR has become a global threat to human and animal health. Development of resistance can be the result of both chromosomal mutations and the acquisition of mobile genetic elements (MGEs), harboring AMR gene mutations [2,3]. It has been reported that, antibiotics are no longer effective against infection-causing bacteria due to increased AMR rate, as a result, every 10 min a patient dies in the USA or Europe [4,5]. However, a substantially higher prevalence of increased AMR is likely to be found in developing countries especially in Africa and Asia due to limited diagnosis facilities, unauthorized antibiotics sale, poor patient education, the inappropriate function of drug regulatory action, inappropriate prescription practices, and non-human practice of antibiotics in livestock sectors [6,7]. Due to the magnitude of the threat, the World Health Organization (WHO) recommended global surveillance programs for animal and human populations.

According to WHO, the first-ever list of antibiotic-resistant “priority pathogens”, *Escherichia coli* is included in the most critical group of all twelve families of bacteria that carriage the greatest threat to human and animal health [8]. The level of antimicrobial resistance in *E. coli* has been used as an indicator of resistance dissemination in bacterial populations, and of selective pressure imposed by antimicrobials used in food animals and humans [9,10,11,12]. However, the frequency of AMR in *E. coli* depends on the source of the isolates. Animal origin has been reported to be the cause of drug-resistant *E. coli* infections in humans, and that these agents harbored the same mobile resistance genes found in diverse bacterial species from a variety of animal sources [13,14,15,16,17]. However, a high prevalence of AMR *E. coli* was isolated from chicken compared with other animals’ origins [18]. Additionally, AMR *E. coli* isolated from humans is similar to *E. coli* from poultry [19]. It has been reported that commensal *E. coli*, can serve as a good reservoir of resistance genes with the ability to transfer these genes to pathogens in the hosts as well as in the human intestinal tract after the consumption of contaminated foods of animal origin [20]. Furthermore, a number of studies have established the transfer of AMR between commensal bacteria and zoonotic pathogens in various ecological environments [21,22,23]. 

Poultry meat production has been increased and doubled over the past 20 years. Poultry is traded at live bird markets, and products are sold unprocessed with bigger clusters of them in city areas which presents significant public and poultry health challenges. A number of companies have already integrated their operations. Poultry and meat processing is a very new movement in the food processing industry in Bangladesh. It has been said that frozen chickens are mostly obtainable through high-end supermarkets charging premium prices and this market is growing every year. Another market segment is food preparation for the main fast-food chains. The local frozen food market is also growing, at a rate of almost 30% in 2011–2012 over the preceding year. City dwellers, are progressively becoming more conscious of their accessibility and the lifestyle they permit, as they desire to go to supermarket instead of to wet markets to buy their everyday stuff, including frozen chicken meat [24]. Tenants in the city becoming more conscious of their accessibility to safe food. Nonetheless, warranting the microbiological safety of frozen chicken meat evolves as a challenge.

Few studies have already reported bacterial contamination in frozen chicken meat from different cities in Bangladesh. Customers in cities have a habit of buying frozen chicken meat along with other frozen and ready-to-cook foodstuffs as these frozen items need slight processing for cooking and, thus, they can save time [25,26,27]. Two of these studies were bacteriological along with AMR phenotype. Another study includes a few genes related to extended spectrum beta-lactamase (ESBL) and non- ESBL producing *E. coli*. However, recent reviews reported the uses of nineteen and ten different types of antibiotics in the broiler and layer farms, respectively in Bangladesh [28]. Therefore, further study is needed for genotyping which shows higher diversity than phenotypes and consequently allows for more accurate comparisons between resistant bacterial populations [29,30]. The aim of this study was to determine the prevalence of *E. coli* in frozen chicken and phenotypic AMR profile as well as the detection of ARG. 

## 2. Results 

### 2.1. Prevalence of E. coli in Frozen Chicken Meat 

*E. coli* was isolated and identified in fifty samples out of 160 samples with an overall prevalence of 31.25%. However, the prevalence of *E. coli* was found to range from 20.0–40.0% in the tested samples (Table 1). The highest prevalence (40.0%) was found in chickens purchased from supermarkets in the Mirpur region while the lowest prevalence (20.0%) was in samples of the Gulshan region.

### 2.2. Antimicrobial Resistance Profiles of E. coli 

Antimicrobial resistance among *E. coli* isolates was determined by the disc diffusion method using seven different antibiotics spanning six different classes. The distribution of AMR patterns is presented in Table 2 and Figure 1.

Overall, 35.7, 16.0 and 48.3% of the isolates were found resistant, intermediate, and sensitive, respectively to all the antibiotics used in this study. About 2.0% of isolates showed resistance to seven antibiotics spanning six classes of antimicrobial agents (Str-Gen-Tet-Amp-Ery-Chl-Sul) ( Appendix A).

About 90.0% of the isolates were found to MDR and about 52.0% of the isolates showed resistance against 4–7 different antimicrobials (Table 3). 

### 2.3. Antimicrobial Resistance Genes (ARGs) in E. coli 

Ten ARGs were detected using PCR in all isolated *E. coli* and the results are presented in Table 5. Tetracycline efflux genes *tet(A)* and *tet(B)* were found in 66.0 % and 60.0 % of the total isolates in this study, respectively. Both *tet(A)* and *tet(B)* genes were found in 60.0% of the isolates. About 64.0% of the isolates harbored the erythromycin esterase (*ereA*) gene. Besides, 56.0 %, 56.0 %, 44.0% of the isolates were found to carrying *aadA1*, *sul1*, *aac(3)-IV* genes, respectively. The presence of the AmpC beta-lactamase-producing gene (*bla_CITM_*) was observed in 48.0% of chicken *E. coli* isolates. Moreover, about 40.0% of chicken *E. coli* isolates carried genes coding for extended-spectrum SHV (*bla_SHV_*) beta-lactamases (Table 4). 

Antimicrobial resistance genes (3-9) were detected in 84.0% of the isolates. While 70.0% of the isolates were found to carry 5–9 of the ten ARGs investigated in this study (Figure 2). 

### 2.4. Antimicrobial Resistance Phenotype and Genotype Association 

Strong positive associations were found among AMR phenotypes and the corresponding resistance genes except for tetracycline-*tet(B)* (OR: 1.33, 95% CI 0.41–4.31, *p* = 0.63) (Table 5). The observed strongest associations were between the following pairs of antibiotics and corresponding genes: tetracycline-*tet(A)* (OR: 512.0, 95% CI: 30.03–8728.99, *p* ≤ 0.0001), streptomycin-*aadA1* (OD: 270.0, 95% CI: 22.86–3189.39, *p* < 0.0001), erythromycin-*ere(A)* (OD: 255, 95% CI: 21.50–3024.21, *p* ≤ 0.0001), sulfonamide-*sulI* (OD: 82.33, 95% CI: 12.51–542.00, *p* ≤ 0.0001), gentamicin-*aac(3)-IV* (OD: 52.0, 95% CI: 9.57–291.19, *p* ≤ 0.0001). Positive associations were also observed for other antibiotics and corresponding genes analyzed in this study. By pairwise association analysis, non-significant positive and negative associations were found within AMR phenotypes and genotypes ( Appendix A).

## 3. Discussion

*E. coli* is recognized as a common inhabitant of the vertebrate intestinal tract which frequently causes contamination in retail meat products. It is one of the most common food-borne pathogens associated with mortality in commercial poultry as well as condemning the carcasses in slaughterhouses and has been considered a significant public health threat and economic burden [31]. It has been reported that resistant strains from the gut readily contaminate poultry carcasses at slaughter, and consequently, poultry meats are often contaminated with resistant *E. coli* [32]. Antibiotics have been widely used for preventing economic losses caused by *E. coli* and increasing production efficiency [33]. However, with increased consumption of these drugs may lead to scattering them into manure and other poultry wastes and transferring them to humans by their residues in carcasses and can be the origin of bacterial resistance, mortality, and increase in human hospitalization [34]. In this study, the overall prevalence of *E. coli* in frozen chicken was found 31.25% which is lower than the prevalence (76.1%) reported from frozen chicken [25]. Our findings are also lower than the findings of the previous study [35]. It has been reported about a 63.5% prevalence of *E. coli* in raw chicken meat covering both layer and broiler swab samples. We have taken about 10 g of meat from the surface of the breast and thigh muscles of each of the broilers. However, processed meat samples for *E. coli* isolation from various parts of the body of layer, broiler and cockerel has been examined [25,35]. On the other hand, the present study was limited only to Dhaka city. Moreover, sampling time, season, etc. were also different. All of these factors may contribute to the differences of *E. coli* prevalence in frozen meat samples. Moreover, broilers sold in supermarkets especially come from contract farms that manage their farms more hygienically than the general farmers may also contribute to lower occurrences of *E. coli*. A contract farm is defined as a farm where farmers have a contract with the company (supermarket authority) that the company provides the chicks, the feed, veterinary care, and technical advice, etc. while the farmers provide the day-to-day care of the birds, land, and housing, as well as utilities/maintenance of the housing and finally share benefits as per contract [25]. Additionally, the prevalence may not show actual prevalence as we have examined a portion of muscle sample from the surface of the frozen chicken. The source of *E. coli* may be the chicken itself or it comes from contamination during the dressing and packing of chicken. It is to be mentioned that we have ensured the aseptic handling of samples in the laboratory to avoid laboratory-acquired contamination.

From Bangladesh, many studies have been reported on AMR and the majority of them concentrate on the isolation and investigation of the antibiotic resistance patterns of *E. coli* by disc diffusion technique [36,37]. Although the conventional method is most widely used for determining AMR because of convenience, efficiency, and cost; it has some limitations. Results may be unexpected or borderline in addition to some other limitations such as its inapplicability to many fastidious organisms and anaerobes [38], unable to obtain minimal inhibitory concentration (MIC) values [39], labor-intensive and time-consuming [40]. In this study, we have used both phenotypic detection of AMR as well as detection of ARGs from the same isolates. It has been reported that tetracycline resistance *E. coli* was found more frequently [28]. Besides, resistances were also found against almost all antibiotics used in this study. ESBL *E. coli* isolates from frozen meat displayed resistance to oxytetracycline and amoxicillin (91.9%), ampicillin and trimethoprim–sulfamethoxazole (89.2%), pefloxacin (87.8%), cefepime (81.1%), piperacillin–tazobactam (73.0%), and doxycycline (70.3%) [25]. A recent review [28] reported that nineteen and ten different types of antibiotics are used in the broiler and layer farms, respectively in Bangladesh. The most commonly used antibiotics included ciprofloxacin, ampicillin, amoxicillin, trimethoprim, oxytetracycline, tylosin tartrate, tiamulin, norfloxacin, enrofloxacin, doxycycline, and colistin sulfate. Information regarding the use of antimicrobials in broiler was not available to the research team to draw further insights.

MDR bacteria are an emerging clinical challenge in the poultry sector as well as the livestock sector. In this study, about 90.0% of the *E. coli* isolates were found MDR, and 52% of the isolates showed resistance against 4–7 different antimicrobials. Our findings are within the findings of recent reports regarding MDR phenotypes of *E. coli*. It has been reported eighty-six *E. coli* isolates from frozen chicken meat against sixteen antimicrobials and found that all the isolates are MDR [25] and as suggested by other literature reported 49.23 and 51.09% MDR *E. coli* isolates from broiler and layer meat samples [35]. AMR pattern (streptomycin-gentamicin-tetracycline-ampicillin-erythromycin-chloramphenicol-sulfonamide) of one *E. coli* isolates ( Appendix A) indicates the necessity of prudent use of antibiotics. AmpC beta-lactamase-producing gene (*bla_CITM_*) and the gene coding for extended-spectrum SHV beta-lactamases (*bla _SHV_*) were detected in broiler chicken *E. coli* isolates in the present study. It has been also reported that 12.8% of broiler chicken *E. coli* isolates carried *bla_SHV_* and 4.56% of isolates possess *bla_CITM_* genes [35]. Differences in findings might reflect the sources and number of samples etc. In Bangladesh, *blaCTX-M-1* (94.4%) and *bla_TEM_* (50–91.3%) ESBL-producing *E. coli* were reported in the droppings of chickens [41,42]. Strong correlations between most of the antimicrobial-resistant phenotypes and genotypes were observed among the investigated *E. coli* isolates that the similar findings are reported earlier [35].

In *E. coli*, the AMR phenotypic-genotypic agreement of 33–85% [29] has been reported for different antimicrobial agents and related genes. In the present study, it was found that few isolates with resistance phenotypes lacked the corresponding ARGs tested, indicating the occurrence of multi-gene mediated AMR. On the other hand, some isolates carry the resistance genes but phenotypically not resistant to the corresponding antibiotics used in this study. The occurrence of similar AMR phenomena was also reported previously [29]. Sometimes, the phenotype or the genotype alone is unable to accurately predict the outcome of the other, as molecular mechanisms of AMR are multifaceted. Thus, the presence or absence of a specific gene corresponding to a particular phenotype does not necessarily infer that the particular strain is resistant or susceptible [43]. The differences between the genotype and phenotype observed in this study might be due to not testing for all possible resistance genes, or genes not being turned on, or the presence of ‘silent gene cassettes’ in certain isolates. 

It is established that the use of a specific antibiotic can result in its own resistance. It can also play a role as a co-selection marker for other antibiotics. This may happen in completely unrelated drug classes [44,45]. The use of chloramphenicol in the poultry sector is very rare. However, about 22.0% of the isolates showed resistance to chloramphenicol. Moreover, chloramphenicol resistance genes viz. *catA1* and *cmlA* were detected in 36.0 and 34.0% of the *E. coli* isolates, respectively. Resistance to chloramphenicol might be due to the co-selection dynamics among chloramphenicol, oxytetracycline, and sulfonamide [30,44]. A non-significant poor association between *tet(A)* and *tet(B)* resistance genes among *E. coli* isolates (Appendix A) was observed which may be due to the incompatibility of plasmids carrying the tetracycline resistance determinants [30]. However, further study is required to enumerate the relationships among the resistance gene(s) and the probable link to antimicrobial exposure.

The findings of this study indicated that more caution are required for personnel hygiene in the processing and handling of poultry and poultry products to prevent the transfer of AMR *E. coli* from frozen poultry sold in supermarkets in Bangladesh. Present findings also highlighted the necessity of cautious use of antimicrobials in chickens to minimize the development of antibiotic-resistant bacterial strains. The study has limitations and these include a small sample size and fewer antibiotic-resistance genes tested. Further detailed investigation using a large number of samples, targeting more antibiotics including latest antibiotics as well as more ARGs, etc. would provide broader insights into the AMR patterns, prevalent ARGs, etc. among clinically important pathogens from food producing animals. 

## 4. Materials and Methods

### 4.1. Collection of Whole Frozen Chicken

This cross-sectional study was conducted in Dhaka district in Bangladesh. A total of 160 frozen chicken meat were purchased from different supermarkets located in Gulshan, Dhanmondi, Mirpur, and Uttara regions in Dhaka city (Figure 3) during the period of July 2018 to June 2019 for the isolation and identification of *E. coli*. After purchase chicken samples were individually placed in a sterile zipper bag, kept in an ice box, and immediately brought to the laboratory of the Animal Biotechnology Division, National Institute of Biotechnology. Samples were either shortly stored in the refrigerator (4 °C) in case of immediate processing or at −20 °C in case of processing after 1–2 days of purchase. 

### 4.2. Sample Processing and Isolation of E. coli

The preparation of the meat samples was based on the slight modification of the European standard ISO-16654:2001 [46] About 10 g of meat sample, thigh and breast muscles each 5 g, was obtained from the surface of each of the chickens, cut into small pieces, added with 90 ml of sterile 1% peptone and mixed well. Enrichment was performed for 16 to 24 h at 150 rpm at 37 °C in a shaking incubator. A portion of enriched samples (25 µL) were plated on MacConkey’s agar (MCA; Difco) and incubated at 37 °C for 24 h. Typical colonies of *E. coli* were randomly picked, mixed with 100 µL phosphate-buffer saline, inoculated onto eosine methyline blue (EMB) agar, and incubated at 37 °C for 18–24 h. After incubation, the selected bacterial colonies from EMB agar were inoculated into 5 mL of sterile Luria Bertani (LB) broth and placed into a shaking incubator at 37 °C for overnight. This culture was used for further analysis. 

### 4.3. Identification of E. coli by Polymerase Chain Reaction (PCR)

Genomic DNA was isolated from selected bacteria cultured in LB broth by using a mixture of phenol: chloroform: isoamyl alcohol (25:24:1) [47], followed by precipitation with isopropanol. Finally, the DNA was dissolved in 50 µL of Tris-EDTA buffer. The concentration (ng/µL) and absorbance ratio (260 nm/280 nm) was determined by spectrophotometry (NanoDrop 2000, Thermo Scientific). PCR amplification was performed using primers (16E1-F: 5′-GGGAGTAAAGTTAATCCTTTGCTC-3′ and 16E2-R: 5′-TTCCCGAAGGCACATTCT-3′) targeting 584 bp fragment of 16S rRNA gene as reported earlier [48]. PCR amplification was performed on 25 μL scale, containing 1.5 mM MgCl_2_, 50 mM KCl, 10 mM Tris-HCl (pH 9.0), 0.1% Triton X-100, 200 μM of each dNTP, 1 μM primers, 1 unit of Taq DNA polymerase, and 5 μL (~50 ng/μL) of genomic DNA in Gene Atlas thermocycler (ASTEC Gene Atlas, G02, Japan). The thermal condition included an initial denaturation for 5 m at 95 °C, followed by 35 cycles consisting of denaturation at 94 °C for 1 m, annealing at 55 °C for 90 s and extension at 72 °C for 1 m, and a final extension of 10 m at 72 °C. Amplified DNA was separated on 1.5% agarose gel and visualized under ultraviolet light in an Axygen™ Gel documentation system (Corning Inc., Corning, NY, USA).

### 4.4. Antimicrobial Resistance Profiling (ARP) of the Isolates 

AMR profiling of the *E. coli* was performed by disc diffusion method according to Clinical and Laboratory Standards Institute (CLSI) guidelines, 2018 [49]. A total of seven antimicrobials from six different classes were used in the AMR profile test. These included (i) aminoglycosides: gentamicin (10 µg) and streptomycin (10 µg), (ii) tetracyclines: tetracycline (30 µg), (iii) β-lactam: ampicillin (10 µg), (iv) macrolides: erythromycin (15 µg) (v) phenicols: chloramphenicol (30 µg), and (vi) sulfonamides: sulfonamide (300 µg) (oxoid^TM^). *E. coli* (ATCC 25922) strain was used as a reference strain for interpretations of the antimicrobial susceptibility test results. The isolates were categorized as resistant, intermediate, or sensitive based on the diameter of the zone of inhibition according to CLSI guidelines. As there is no standard zone of inhibition mentioned for erythromycin with *Enterobacteriaceae,* the interpretation was performed based on the zone of inhibition for *Staphylococcus* spp. *E. coli* showed resistance to three or more than three different classes of antimicrobials, which was defined as multidrug-resistance (MDR) [50]. 

### 4.5. Detection of Antimicrobial Resistance Genes (ARGs) in the Isolates

All the isolates were tested for the presence of *aadA1, aac(3)-IV, tet(A), tet(B), bla_SHV_, bla_CITM,_ ereA, catA1, cmlA,* and *sulI* ARGs by PCR as described [51,52,53]. Details of the primer sequences, annealing temperature, PCR product size, etc. are presented in Table 6. Basic thermal conditions were initial denaturation for 5 m at 95 °C, 35 cycles consisting of denaturation for 1 m at 94 °C, annealing for 40 s at the temperature of each respective gene and extension for 1 m at 72 °C, followed by a final extension step of 10 m at 72 °C. The annealing temperature varied for each gene (Table 6). 

### 4.6. Statistical Analysis

Descriptive and association-based statistical analyses were conducted using Microsoft Excel v.13.0 and GraphPad Prism v.8.0 statistical tools, respectively. The association between specific AMR phenotype and the ARG was calculated and an association was considered significant at a p-value of <0.05 and was reported as an odds ratio (OR) with 95% confidence intervals (CI). An OR of > 1 was considered a positive association or the increasing probability of the co-occurrence of the genotype or phenotype, while an OR of <1 was considered a negative association or the decreasing probability of the co-occurrence of the genotype or phenotype. The degree of agreement between phenotypic and genotypic relations was assessed by Kappa coefficients (κ) [54]. 

## 5. Conclusions

The results of the present study indicate that a good percentage of frozen chicken sold in supermarkets in Dhaka city carries *E. coli* and they are resistant to commonly used antibiotics and the majority of them are MDR. Furthermore, more cautions are necessary for choosing a drug for the treatment of clinical cases of poultry because the transfer of drug resistant gene from one bacterium to other may be hazardous for human being too. Therefore, careful use of antimicrobials in poultry production is recommended.

## Figures and Tables

**Figure 1 antibiotics-12-00041-f001:**
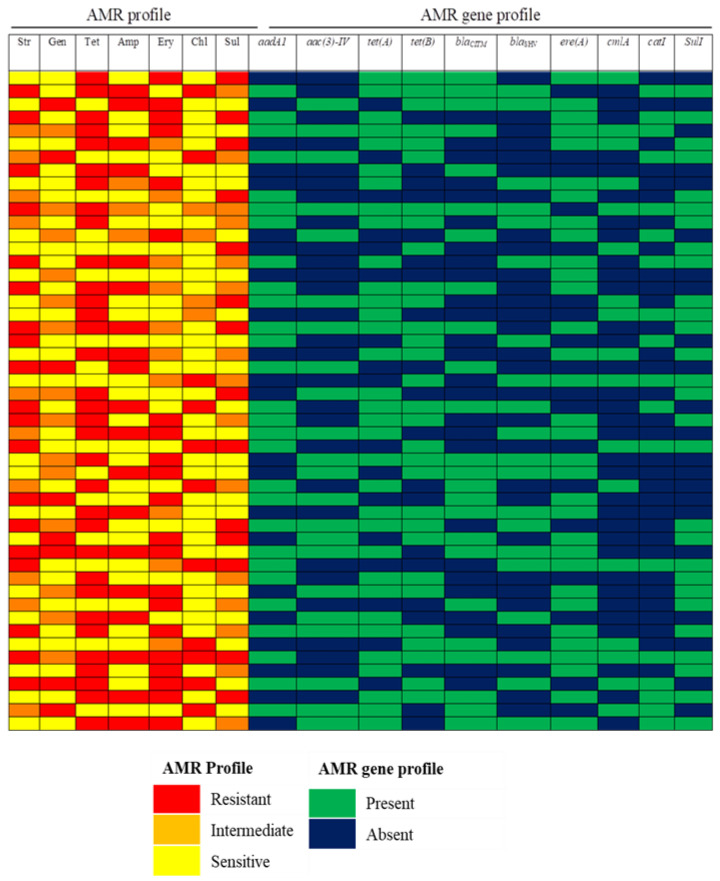
Distribution of antimicrobial resistance phenotype and antimicrobial resistance genes of *E. coli* isolated from chicken meat from supermarkets of Dhaka city. Str: streptomycin, Gen: gentamicin, Tet: tetracycline, Amp: ampicillin, erythromycin, Chl: chloramphenicol, Sul: sulfonamide.

**Figure 2 antibiotics-12-00041-f002:**
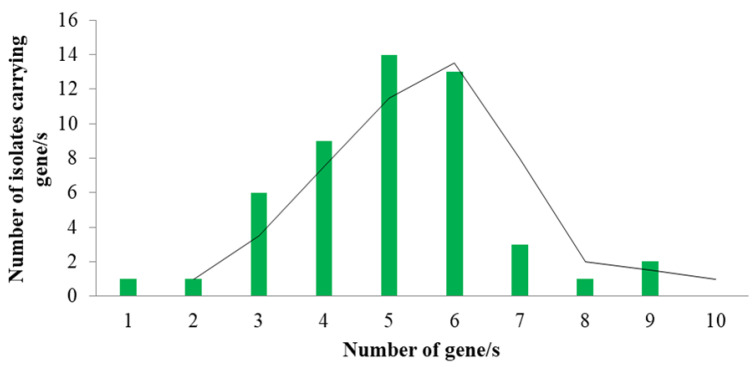
A number of antimicrobial resistance genes (ARGs) detected in isolated *E. coli* from chicken meat from supermarkets in Dhaka city (*n* = 50).

**Figure 3 antibiotics-12-00041-f003:**
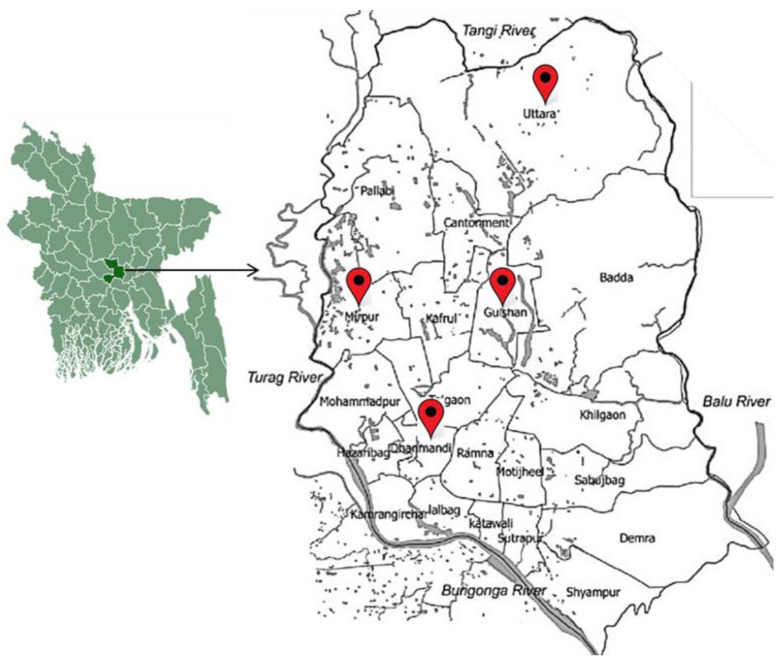
Sample collection from supermarkets of four different locations in Dhaka city, Bangladesh.

**Table 1 antibiotics-12-00041-t001:** Prevalence of *E. coli* in frozen chicken meat collected from different supermarkets located in Dhaka city.

Location	Number of Supermarkets	No. of Chicken Sample	Sources of Chicken	No. of Positive Sample	Prevalence (%)
Gulshan	4	40	Contract farmers	8	20.0
Dhanmondi	4	40	Contract farmers	14	35.0
Mirpur	4	40	Contract farmers	16	40.0
Uttara	4	40	Contract farmers	12	30.0
Overall		160		50	31.25

**Table 2 antibiotics-12-00041-t002:** Antimicrobial resistance profile of *E. coli* isolated from frozen chicken meat (*n* = 50).

Antimicrobial Class	Antimicrobial Agent (Conc.)	No. of *E. coli* Tested	No. of Isolates (%)		
			Resistance	Intermediate	Sensitive
Aminoglycosides	Streptomycin (10 µg)	50	19 (38.0)	10 (20.0)	21 (42.0)
	Gentamicin (10 µg)	50	8 (16.0)	14 (28.0)	28 (56.0)
Tetracyclines	Tetracycline (30 µg)	50	33 (66.0)	0 (0.0)	17 (34.0)
Beta-lactams	Ampicillin (10 µg)	50	19 (38.0)	3 (6.0)	28 (56.0)
Macrolides	Erythromycin (15 µg)	50	21 (42.0)	10 (20.0)	19 (38.0)
Phenicols	Chloramphenicol (30 µg)	50	11 (22.0)	4 (8.0)	35 (70.0)
Sulfonamides	Sulfonamide (300 µg)	50	14 (28.0)	15 (30.0)	21 (42.0)
Overall		350	125 (35.7)	56 (16.0)	169 (48.3)

**Table 3 antibiotics-12-00041-t003:** Distribution of resistance profiles of *E. coli* (*n* = 50).

Antibiotic Class	No. of Antimicrobials	No. of Isolate Resistant (%)	MDR ^a^ No. of Isolate (%)
1	1	3 (6.00)	No
2	2	2 (4.00)	
	3	3 (6.00)	
3	3	16 (32.00)	Yes
	4	4 (8.00)	45 (90.00)
4	4	13 (26.00)	
	5	3 (6.00)	
5	5	3 (6.00)	
	6	2 (4.00)	
6	7	1 (2.00)	

^a^ Isolate is defined as multidrug-resistant when it shows resistance to >3 classes of antimicrobial agents.

**Table 4 antibiotics-12-00041-t004:** Distribution of antimicrobial resistance genes (ARGs) in *E. coli* isolates (*n* = 50).

Antimicrobial Class	Antimicrobial Agent	ARGs	No. of *E. coli* Positive (%)	No. of *E. coli* Negative (%)
Aminoglycosides	Streptomycin	*aadA1*	28 (56.0)	22 (44.0)
	Gentamicin	*aac(3)-IV*	22 (44.0)	28 (56.0)
Tetracyclines	Tetracycline	*tet(A)*	33 (66.0)	17 (34.0)
		*tet(B)*	30 (60.0)	20 (40.0)
Beta-lactams	Ampicillin	*bla_CITM_*	24 (48.0)	26 (52.0)
		*bla_SHV_*	20 (40.0)	30 (60.0)
Macrolides	Erythromycin	*ereA*	32 (64.0)	18 (36.0)
Phenicols	Chloramphenicol	*cmlA*	17 (34.0)	33 (66.0)
		*cat1*	18 (36.0)	32 (64.0)
Sulfonamides	Sulfonamide	*sul1*	28 (56.0)	22 (44.0)

**Table 5 antibiotics-12-00041-t005:** Comparison of AMR in *E. coli* isolates according to phenotypic and genotypic results.

Antimicrobial	NP	ARG	NG	P+/G+	P+/G-	P-/G+	P-/G-	Odds Ratio	95% CI	*p*
Streptomycin	29	*aadA1*	28	27	2	1	20	270.00	22.86–3189.39	<0.0001
Gentamycin	22	*aac(3)-IV*	22	19	3	3	25	52.78	9.57–291.19	<0.0001
Tetracycline	33	*tet(A)*	33	32	1	1	16	512.00	30.03–8728.99	<0.0001
		*tet(B)*	30	20	12	10	8	1.33	0.41–4.31	0.63
Ampicillin	22	*bla_CITM_*	24	15	7	9	19	4.52	1.37–14.98	0.01
		*bla_SHV_*	20	13	9	7	21	4.33	1.30–14.47	0.02
Erythromycin	31	*ereA*	32	30	1	2	17	255.00	21.50–3024.21	<0.0001
Chloramphenicol	15	*cmlA*	17	11	4	6	29	13.29	3.14–56.27	0.0004
		*cat1*	18	12	3	6	29	19.33	4.14–90.24	0.0002
Sulfonamide	29	*sulI*	28	26	3	2	19	82.33	12.51–542.00	<0.0001

NP: number of *E. coli* isolates expressing phenotypic resistance; ARG: antibiotic resistance gene; NG: number of *E. coli* isolates carrying the indicated resistance gene; P+/G+: number of phenotypically resistant *E. coli* isolates (P+) with resistance gene (G+) for the drug identified; P+/G-: number of phenotypically resistant *E. coli* isolates (P+) with no resistance gene (G-) for the drug identified; P-/G+: number of phenotypically susceptible *E. coli* isolates (P-) with resistance gene (G-) for the drug identified; P-/G-: number of phenotypically susceptible *E. coli* isolates (P-) with no resistance gene (G-) for the drug identified; CI: confidence interval.

**Table 6 antibiotics-12-00041-t006:** Primers used for amplification of antimicrobial resistance genes (ARGs) from *E. coli*.

Antimicrobial Agent	Resistance Gene	Sequence (5″-3″)	Size (bp)	Annealing Temp (°C)	References
Streptomycin	Adenylyl transferases (*aadA1*)	F- TATCCAGCTAAGCGCGAACT	447	58	[51]
		R- ATTTGCCGACTACCTTGGTC			
Gentamicin	Aminoglycoside acetyltransferases (*aac(3)-IV*)	F- CTTCAGGATGGCAAGTTGGT	286	55	[52]
		R- TCATCTCGTTCTCCGCTCAT			
Tetracycline	Efflux pump resistance (*tet(A)*)	F- GGTTCACTCGAACGACGTCA	577	57	[51]
		R- CTGTCCGACAAGTTGCATGA			
	Efflux pump resistance (*tet(B)*)	F -CCTCAGCTTCTCAACGCGTG	634	56	
		R- GCACCTTGCTGATGACTCTT			
Ampicillin	*β*-lactamase encoding penicillin resistance (*Bla_SHV_)*	F- TCGCCTGTGTATTATCTCCC	768	52	[52]
		R- CGCAGATAAATCACCACAATG			
	*β*-lactamase encoding cephalosporin resistance (*Bla_CITM_*)	F- TGGCCAGAACTGACAGGCAAA	462	47	
		R- TTTCTCCTGAACGTGGCTGGC			
Erythromycin	Erythromycin esterase (*ereA*)	F- GCCGGTGCTCATGAACTTGAG	419	60	
		R- CGACTCTATTCGATCAGAGGC			
Chloramphenicol	Acetyltransferases (*catA1*)	F-AGTTGCTCAATGTACCTATAACC	547	55	
		R- TTGTAATTCATTAAGCATTCTGCC			
	Transporter resistance (*cmlA*)	F- CCGCCACGGTGTTGTTGTTATC	698	33	
		R- CACCTTGCCTGCCCATCATTAG			
Sulfonamide	Dihydropteroate synthase (*sul1*)	F- TTCGGCATTCTGAATCTCAC	822	47	[53]
		R- ATGATCTAACCCTCGGTCTC			

## Data Availability

Not applicable.

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
