# Peer review of "Isolation, Identification and Genetic Characterization of Antibiotic Resistant Escherichia coli from Frozen Chicken Meat Obtained from Supermarkets at Dhaka City in Bangladesh"

_antibiotics, 2022, doi:10.3390/antibiotics12010041_

Round 1

Reviewer 1 Report

Kamal et al., entitled"Isolation, identification and genetic characterization of antimicrobial resistant Escherichia coli from frozen broilers obtained from supermarkets at Dhaka city in Bangladesh" has focused on AMR in frozen broiler meat. The manuscript is written well. However, the authors targeted only on small sample size and only limited to E. coli and a few ARGs. The authors missed the prevalence of AMR in some important drugs like carabapenem, and colistin and also missed some important ARGs such as a blaCTX-M group of ESBLs. It would be better if the authors also identify some virulence genes. 

The following are some minor and major concerns that need to be addressed.

Results: 

Section 2.1. Prevalence of E. coli in frozen broiler: Line 92-96: "A total of 160 frozen broilers ................... the isolates tested." These lines are related to your methodology. Please move them to the methodology section.

Line 129: Table 3: The results of the fourth column of Table 3 "MDR isolates" is confusing. please revise it.

Line 166: Table 5: Add footnotes for the abbreviations used in Tables. For example: NP= , NG ,P+G etc..............

Discussion:

Line 220-221: Please correct the word meet into (meat) and resistances into (resistance)

Line 232,241, 268: No need to mention tables in the discussion section. 

Methodology:

Line 349: What were the criteria for selecting only specific ARGs? Some important resistant genes like blaCTX-M in ESBL, and MCR genes for Colistin, and carbapenems are missing.

Line 358: Table 6: References are missing for some primers in the table.

Add some new relevant studies in Introduction and discussion section

DOI: 10.1111/jam.15469 ; 

DOI: 10.3390/antibiotics11111551

DOI: 10.1016/j.micpath.2020.104722

Author Response

Point to point responses

Manuscript ID: Antibiotics- 2096799

Comments and Suggestions from the reviewers:

(Highlighted in yellow in the revised MS according to reviewer’s comments).

Reviewer#1:

Kamal et al., entitled "Isolation, identification and genetic characterization of antimicrobial resistant Escherichia coli from frozen broilers obtained from supermarkets at Dhaka city in Bangladesh" has focused on AMR in frozen broiler meat. The manuscript is written well. However, the authors targeted only on small sample size and only limited to E. coli and a few ARGs. The authors missed the prevalence of AMR in some important drugs like carabapenem, and colistin and also missed some important ARGs such as a blaCTX-M group of ESBLs. It would be better if the authors also identify some virulence genes.

Results:

Section 2.1. Prevalence of E. coli in frozen broiler: Line 92-96: "A total of 160 frozen broilers ................... the isolates tested." These lines are related to your methodology. Please move them to the methodology section.

Reply: Thank you for your suggestion. It has been revised. See the line no.308-311.

Line 129: Table 3: The results of the fourth column of Table 3 "MDR isolates" is confusing. please revise it.

Reply: Thank you for your comments. It has been revised. See the line no.135.

Line 166: Table 5: Add footnotes for the abbreviations used in Tables. For example: NP= , NG ,P+G etc..............

Reply: Thank you for pointing out. It has been revised. See the line no. 184-189.

Discussion:

Line 220-221: Please correct the word meet into (meat) and resistances into (resistance)

Reply: Thank you for pointing out. It has been revised. See the line no. 238

Line 232,241, 268: No need to mention tables in the discussion section. 

Reply: Thank you for pointing out. It has been revised of mentioned areas.

Methodology:

Line 349: What were the criteria for selecting only specific ARGs? Some important resistant genes like blaCTX-M in ESBL, and MCR genes for Colistin, and carbapenems are missing.

Reply: Thank you for pointing out the matter. Actually genes are selected randomly as well as available of primers in the laboratory. This study has been done during the period of 2018-2019.  Among others we have used blaCITM and blaSHV. We will consider your comments in our future studies.

Line 358: Table 6: References are missing for some primers in the table.

Reply: Thank you for pointing out the matter. It has been added. See the table line no. 380

Add some new relevant studies in Introduction and discussion section

Reply: Thank you for pointing out. It has been added.

Reviewer 2 Report

Comments to authors

The manuscript entitled “Isolation, identification and genetic characterization of antimicrobial resistant Escherichia coli from frozen broilers obtained from supermarkets in Dhaka city in Bangladesh” described the isolation of E. coli from frozen chicken meat collected from supermarkets located in Dhaka city in Bangladesh. The purpose of this study is interesting. However, some flaws should be revised. I also recommend checking the English writing style and grammar with English professionals.

 Title

1.     It might be better if the authors change “frozen broiler” to “frozen chicken meat” or “frozen broiler meat” (revise all points in the whole manuscript).

2.     Change “antimicrobial” to “antibiotic”

 Introduction

1.     Line 38: humans

2.     Line 54: italicize “Escherichia coli

3.     I found many studies on AMR E. coli isolated from chicken meat from several reports in Bangladesh. Authors should add the novelty of this study when compared with the previous reports. This point should be added to the manuscript.

 Discussion

1.     The maximum acceptable level and detection limit of E. coli in frozen chicken meat by Bangladesh’s food regulation or global food authorities should be discussed and compared with your result.

 Results

1. Lines 92-96 should be removed. It was the summary of the study design. Authors should show only the prevalence of E. coli obtained.

2.     Table 1: Please change “super shop” to “supermarkets”.

3.     Table 2: change “08” to “8”, “03” to “3” and “04” to “4”

4.     Could authors show the result of the antibiotic-resistant profile of standard E. coli ATCC25922? Maybe add to section 2.2

  Materials and Methods

1.     Section 4.2: References for this method should be added.

2.     Section 4.2: Is this isolation method under the standard protocol? References should be added if authors worked under the standard protocol.

3.     Section 4.3: Did authors confirm all isolates by DNA sequencing result or comparing the DNA band on gel electrophoresis with DNA of standard strain? The method should be finely described.

Author Response

Point to point responses

Manuscript ID: Antibiotics- 2096799

Comments and Suggestions from the reviewers:

(Highlighted in yellow in the revised MS according to reviewer’s comments).

Reviewer#2

The manuscript entitled “Isolation, identification and genetic characterization of antimicrobial resistant Escherichia coli from frozen broilers obtained from supermarkets in Dhaka city in Bangladesh” described the isolation of E. coli from frozen chicken meat collected from supermarkets located in Dhaka city in Bangladesh. The purpose of this study is interesting. However, some flaws should be revised. I also recommend checking the English writing style and grammar with English professionals

Title:

  1. It might be better if the authors change “frozen broiler” to “frozen chicken meat” or “frozen broiler meat” (revise all points in the whole manuscript).

Reply: Thank you for pointing out the matter. It has been revised.

  1. Change “antimicrobial” to “antibiotic”

Reply: Thank you for pointing out the matter. We have used sulfonamide disc in our research which are not truly antibiotics. As per definition an antibiotic is a substance produced by bacteria or fungi which have antimicrobial activity, whereas sulfonamides are synthetic. For this instance, we have used terminology “antimicrobial” rather than “antibiotics”. Hope the matter will be considered.

Introduction: 

  1. Line 38: humans

Reply: Thank you for pointing out the matter. It has been modified.

  1. Line 54: italicize “Escherichia coli”

Reply: Thank you for pointing out the matter. It has been modified.

  1. I found many studies on AMR E. coli isolated from chicken meat from several reports in Bangladesh. Authors should add the novelty of this study when compared with the previous reports. This point should be added to the manuscript.

Reply: Thank you for pointing out the matter. See the line no. 205-215, 235-243 and 249-266 of discussion section.  

Discussion:

  1. The maximum acceptable level and detection limit of E. coli in frozen chicken meat by Bangladesh’s food regulation or global food authorities should be discussed and compared with your result.

Reply: Thank you for pointing out the matter. In our country we have not yet been set an acceptable and detection limit of E. coli in frozen chicken or any other food stuffs. Our primary objective in this study was to determine the presence of E. coli rather than how many E. coli present in chicken meat sample. We followed the below reference guidelines to compare our findings. We found the range of bacterial load in meat sample in satisfactory level.

Ref: Feng, P., Weagant, S. D., Grant, M. A., Burkhardt, W., Shellfish, M., and Water, B. (2002) Bacteriological Analytical Manual (BAM) Chapter 4: Enumeration of Escherichia coli and the Coliform Bacteria. Bacteriological analytical manual (BAM), US Food and Drug Administration.

Results:

  1. Lines 92-96 should be removed. It was the summary of the study design. Authors should show only the prevalence of E. coli obtained.

Reply: Thank you for pointing out the matter. It has been modified.

  1. Table 1: Please change “super shop” to “supermarkets”.

Reply: Thank you for pointing out the matter. It has been modified.

  1. Table 2: change “08” to “8”, “03” to “3” and “04” to “4”

Reply: Thank you for pointing out the matter. It has been modified.

  1. Could authors show the result of the antibiotic-resistant profile of standard E. coli ATCC25922? Maybe add to section 2.2

Reply: Thank you for pointing out the matter. We strictly follow the Clinical and Laboratory Standards Institute (CLSI) Performance Standards for Antimicrobial Susceptibility Testing. 28th ed. CLSI supplement M100.Wayne, PA: Clinical and Laboratory Standards Institute, 2018.  This part has been added into the Material and Methods section of the manuscript. See the line no 326-327

Materials and Methods:

  1. Section 4.2: References for this method should be added.

Reply: Thank you for pointing out the matter. Reference has been added.

  1. Section 4.2: Is this isolation method under the standard protocol? References should be added if authors worked under the standard protocol.

Reply: Thank you for pointing out the matter. Reference has been added.

Ref -46: WHO. Laboratory protocol. In Isolation of Salmonella spp. From Food and Animal Faeces, 5th ed.; WHO: Geneva, Switzerland, 2010; Volume 13, pp. 4–8.

  1. Section 4.3: Did authors confirm all isolates by DNA sequencing result or comparing the DNA band on gel electrophoresis with DNA of standard strain? The method should be finely described.

Reply: Thank you for pointing out the matter. We have confirmed all E. coli strain by PCR using specific primers as reported and this part have been added into methods. Line 344-345 (Tsen et al., 1998). Besides, we have randomly sequenced some samples to further confirm the isolates.

Round 2

Reviewer 1 Report

The authors have clarified all the concerns and comments. 

Reviewer 2 Report

1.     Reference no. 46 should be referenced from “ISO 16654:2001 Microbiology of food and animal feeding stuffs — Horizontal method for the detection of Escherichia coli O157”. The reference present in Ref. no 46 is for Salmonella isolation, not for E. coli. Kindly check the reference format.

See the relevant documents:

https://www.iso.org/standard/29821.html

https://www.agriculture.gov.au/sites/default/files/sitecollectiondocuments/aqis/exporting/meat/elmer3/approved-methods-manual/E-coli-O157-Microbiology-of-food-and-animal-ISO-16654.2001.pdf

2.     Line 30: Change “multi-drug” to “multidrug.

3.     Line 118-119: Uncapitalize the name of drugs.

4.     Line 123: Repeat “Table 3”

5.     Line 215: Check the mistyping.

6.     Line 339-342: Uncapitalize the name of drugs and drug groups.

I have no comment on the Introduction and Discussion parts. The information that the authors provided is sufficient and structured.

Good Luck!